# Water Treatment Using High Performance Antifouling Ultrafiltration Polyether Sulfone Membranes Incorporated with Activated Carbon

**DOI:** 10.3390/polym14112264

**Published:** 2022-06-01

**Authors:** Zubia Abid, Asad Abbas, Azhar Mahmood, Nosheen Fatima Rana, Sher Jamal Khan, Laurent Duclaux, Kashif Mairaj Deen, Nasir M. Ahmad

**Affiliations:** 1Polymer Research Laboratory, School of Chemical and Material Engineering, National University of Sciences and Technology (NUST), H-12, Islamabad 44000, Pakistan; zubia.phdscme@student.nust.edu.pk (Z.A.); engr.ma.abbas@live.com (A.A.); 2Department of Chemistry, School of Natural Sciences, National University of Sciences and Technology (NUST), H-12, Islamabad 44000, Pakistan; dr.azhar@sns.nust.edu.pk; 3Department of Biomedical Engineering and Sciences, School of Mechanical and Manufacturing Engineering, National University of Sciences and Technology (NUST), H-12, Islamabad 44000, Pakistan; nosheen.fatima@smme.nust.edu.pk; 4Department of Biomedical Engineering and Sciences, School of Interdisciplinary Engineering and Sciences, National University of Sciences and Technology (NUST), H-12, Islamabad 44000, Pakistan; 5Institute of Environmental Sciences and Engineering, School of Civil and Environmental Engineering, National University of Sciences and Technology (NUST), Islamabad 44000, Pakistan; s.jamal@iese.nust.edu.pk; 6Laboratoire Environnement Dynamiques Territoires Montagnes, University of Savoie Mont Blanc, F-73000 Chambéry, France; laurent.duclaux@univ-savoie.fr; 7Department of Materials Engineering, The University of British Columbia, Vancouver, BC V6T 1Z4, Canada; kashifmairaj.deen@ubc.ca

**Keywords:** modified activated carbon, polyether sulfone ultrafiltration membranes, phase inversion method, hydrophilicity, BSA protein, pure water flux, antifouling, antibacterial

## Abstract

Membrane fouling is a continued critical challenge for ultrafiltration membranes performance. In this work, polyether sulfone (PES) ultrafiltration (UF) membranes were fabricated via phase-inversion method by incorporating varying concentrations of APTMS modified activated carbon (mAC). The mAC was thoroughly characterized and the fabricated membranes were studied for their surface morphology, functional groups, contact angle, water retention, swelling (%) porosity, and water flux. The hydrophilicity of mAC membranes also resulted in lower contact angle and higher values of porosity, roughness, water retention as well as water flux. Also, the membranes incorporated with mAC exhibited antibacterial performance against model test strains of gram-negative *Ecoil* and gram-positive *S. aureus*. The antifouling studies based on bovine serum albumin protein (BSA) solution filtration showed that mAC membranes have better BSA flux. The higher flux and antifouling characteristics of the mAC membranes were attributed to the electrostatic repulsion of the BSA protein from the unique functional properties of AC and network structure of APTMS. The novel mAC ultrafiltration membranes developed and studied in present work can provide higher flux and less BSA rejection thus can find antifouling applications for the isolation and concentration of proteins and macromolecules.

## 1. Introduction

Public health is globally threatened due to consumption of polluted drinking water. Availability of clean water is essential for the human population and vital for the sustainability of developing countries. The large fraction of pollutants from industry, sewage and sludge are huge challenge in preserving water sources. Also, microorganisms across the water supply system leads to significant contamination [1]. To date, various carbon materials like graphite, carbon nanotubes, graphene carbon mesosphere and AC are well known for different water treatment applications. Among various carbon materials, AC owe economic viability and availability [2]. The large surface area, porosity and high adsorption capacity makes the AC more effective to be employed in waste water treatment and air pollution control to eliminate several pollutants [3]. AC is produced from multiple agro-resources and consists of graphitic lattice with amorphous solid materials being used in the different forms such as fine powder, granules, fibers, and cloths [4]. The heteroatoms present in the structure of carbon like sulfur, oxygen, phosphorus, nitrogen and hydrogen are generally derived from the starting material or can be introduced later to determine its chemical characteristics [5,6]. The acidic and basic characteristic of carbon surface can be determined by number of delocalized electrons and formation of functional groups by heteroatoms [7].

AC has been used on very large scale for water treatment process but it has been reported that bio-foulants such as bacteria make the disinfectant process difficult due to coagulation at the surface of AC and form biofilm [8]. Thus, if AC possess adsorption and/or antibacterial properties to remove waterborne microorganisms, it would be beneficial for water treatment processes [9]. Different nature of interactions including van der Waals forces, steric, π-π as well as bonding of hydrogen, hydrophobic and electrostatic repulsion/attraction are responsible for biofoulants repulsion/adhesion and growth to the surface of activated carbon [10]. Despite AC is well-known material for many decades to treat contaminated water, however, novel opportunities are continued to generate interest to significantly improve carbon surface by various treatment methods to enhance its potential towards particular contaminants removal from water [11]. Different methods including activation conditions, treatment with additives and precursor can be employed to change chemical and physical structure of AC in literature [12]. Numerous surface modification techniques have been published that explain the mechanism and chemistry of AC surface for the removal of pollutants [13,14,15,16]. These modifications can be chemical, physical and biological.

Contaminated water treatment by using polymer membrane technology have number of advantages like low energy consumption, no phase change, high speed and lesser carbon footprint [17]. Among various polymers employed to fabricate membranes, currently PES has been used as the key polymer for fabrication of UF membranes for domestic and commercial applications of water purification [18]. PES membranes have exceptional chemical/thermal stability and mechanical strength [19]. But the main drawback associated with PES, which restricts its uses as membrane is its hydrophobic nature since the hydrophobicity of PES membrane results in severe fouling problems [20]. The fouling of membrane can be inhibited by enhancing surface hydrophilicity, optimizing surface treatment and porosity as well as electrostatics charges of the membrane [21]. Therefore, studies have intended to develop PES membranes with optimized porosity possess better antifouling properties by incorporating different materials, functional groups on to the membrane, adsorbent and biocidal agents like carbon based materials, zinc oxide, and silver nanoparticles to increase its hydrophilicity [22,23,24,25].

Hydroxyapatite-decorated AC was used as nanofiller in PES UF membranes to reduce fouling and enhance water flux [26]. The fabricated membranes showed 4.6% increase in pure water flux and irreversible fouling was decreased to 6% from 33% for nanocomposite membranes. The adsorption of BSA on the nanocomposite membrane surface was decreased because of electrostatic repulsion. In another study, AC, Chitosan and thiolated chitosan/AC has been seperately incorporated as filler in PES UF membranes to increase its hydrophilicity and reduce fouling [27]. The pure water flux and BSA flux both were high for AC added polymer composite membrane. This is because of the highly porous nature of AC, the protein does not clog within the pores.

In consideration of above, present work described the combined novel approach of blending low to high activated carbon contents into polymer membranes for water treatment. Specifically, antifouling UF PES were fabricated by incorporating varying concentrations of –NH_2_ group mAC for the objective to introduce antifouling characters. Major advantages of these mAC polymer membranes are suppression of aggregation and control the other important characteristics like hydrophilicity and antifouling. The development of membranes by employing low to higher concentration of mAC is described as potential non-adsorbent and antifoulants against model test strains of gram negative *E. coil* and gram positive *S. aureus* in comparison pristine membranes [28]. The antifouling studies also elaborated based on BSA solution filtration through prepared mAC membranes.

## 2. Materials and Methods

### 2.1. Materials

Following reagents were used as received: polyether sulfone (PES) (Sigma-Aldrich, St. Louis, MO, USA), Polyvinylpyrrolidone (PVP) (Sigma-Aldrich), dimethylacetamide (DMAc) (Sigma-Aldrich) 3-Aminopropyltrimthoxysilane (APTMS) (Sigma-Aldrich). Powdered activated carbon (AC) was of Chinese Origin. Nutrient broth (Biolab, Budapest, Hungry), Muller Hinton agar (MHA) (Biolab), Bovine serum albumin protein (BSA) (B005) is acquired from CAISSON Laboratories, Inc. (Smithfield, UT, USA). Deionized water was used throughout the experiment.

### 2.2. Methods

#### 2.2.1. Surface Modification of Activated Carbon (AC)

To modify the surface of AC 15g very fine and dried powder of AC was mixed with 300 mL toluene and 2 mL APTMS solution under magnetic stirring for 24 h at 40 °C under inert environment. Figure 1 represents the schematic of the experimental set-up. Solvent was removed by employing lab rotary evaporator (Scilogex RE-100-Pro, Cormwell Avenue, USA). The resulted APTMS-activated carbon was further washed with toluene and acetone then vacuum dried for 6 h at 60 °C [29]. This dried AC was referred as mAC, consisting of amine functional groups on the surface. The mAC was stored under dried conditions to be used for subsequent experimental work.

#### 2.2.2. Membrane Fabrication

Phase inversion via immersion precipitation technique was adopted to fabricate flat sheet PES UF membranes [30]. The casting solutions consist of PES (18 wt%), PVP (0.2 wt%) and variable amounts of mAC powder (shown in Table 1) in DMAc as solvent. The prepared membranes were designated as pristine membrane MP0 and mAC incorporated membranes as MC1, MC4, MC7 and MC10. Known amount of PES was dissolved slowly into DMAc solvent under stirring to form homogenous solution. PVP and mAC were subsequently added in the PES solution and continuously stirred for next 24 h. The casting solution was then sonicated for 10 min and left for more 3 h for complete removal of air bubbles. Automatic film applicator (filmography electrometer; 20 mms^−1^) was used to cast membrane on a polypropylene fabric (support material). After casting the membranes were immediately immersed into the non-solvent bath containing DI water without evaporation. For complete precipitation and removal of residual DMAc the casted membranes were left in non-solvent bath for 30 min and then membranes were finally dried for overnight at room temperature [31].

### 2.3. Characterization of Activated Carbon

#### 2.3.1. Ash Determination

Ash content in AC was estimated in accordance with ASTM, 2866-70. The combustion was done in a muffle furnace at 650 °C for duration of 2 h followed by weighing the calcination residue.

#### 2.3.2. N_2_ Adsorption-Desorption at 77 K

The adsorption measurement (ASAP 2020, Micromeritics, Norcross, USA) of the AC were carried out at 77 K. Samples were degassed prior to adsorption under N_2_ gas carried out for at least 3h. The specific surface area of the AC was determined by applying the Brunauer, Emmett, and Teller Adsorption (BET) equation (one N_2_ molecule area: 0.162 nm^2^) [32]. The total pore volume was determined by adsorption of liquid volume of nitrogen being adsorbed at a value of 0.99 relative pressure. Furthermore, relative values at P/P_o_ < 0.01 were acquired using constant doses of ~10 cm^3^ g^−1^ at STP, keeping time interval at around 300 s. The bidimensional Non Local Density Functional Theory Model (2D-NLDFT) was applied to calculate pore size distribution (PSD) value. In this model, the adsorption isotherm was applied by assuming a finite slit pores with a diameter-to-width aspect ratio of 6 [33]. The porosity of the ACs were also investigated by adsorption of CO_2_ at the temperature of 273 K. The narrow micropores or ultramicropores pores with size lower than 0.8 nm were also evaluated through CO_2_ adsorption isotherms at 273 K. For this purpose, CO_2_ adsorption was assumed using infinite slit pores model from consideration of diameter of pores less than 0.8 nm [34].

#### 2.3.3. pH_PZC_ and Boehm Titrations

The pH of the point of zero charge (pH_PZC_) of AC was also experimentally calculated through the pH drift method by taking 0.15 g amount of AC in 50 mL 0.01 M NaCl solutions of concentration [35]. The deoxygenation of solutions was done by 1 h continuous N_2_ bubbling to adjust pH of solution to successive initial values between acidic value of 2 and basic value of 12. The suspensions were continuously stirred for next 48 h under N_2_ gas to attain equilibrium pH value. The value of pHP_ZC_ was then estimated by plotting a graph for which initial and final pH values were kept equal. The pH_PZC_ value that controls the acidobasic properties is dependent on the oxygen surface groups’ content. In order to quantify the basic and acidic groups (oxygenated) on the surface of AC, “Boehm” titrations were carried out [36]. The surface functional groups like that of P-containing acidic groups, carboxylic (R-COOH). lactone (R-OCO), carbonyl or quinone (RR′C=O) and phenol (Ar-OH), were determined. For quantification, certain assumptions were made regarding reactivity and non-reactivity. For example, NaOH did not expect to react with the RR′C=O groups, while NaOC_2_H_5_ reacted with all groups present, RR′C=O and R-OH functional groups did not react with Na_2_CO_3_ but R-COOH only reacted with NaHCO_3_ as well as the P-containing acidic groups. Typically, 0.1 M aqueous reactant solution of NaOH or Na_2_CO_3_ or NaHCO_3_ was prepared and 0.05 g of AC was poured into 150 mL. For NaOC_2_H_5_ solution preparation, 0.1 g of AC was taken in a closed polyethylene flask that contained 0.01 mol L^−1^ of 50 mL absolute ethanol. The prepared mixtures were stirred continuously at 150 rpm at 25 °C for 2 days. This was followed by filtration of the suspensions through 0.45 mm Millipore membrane filters. Standardized HCl (0.01 mol L^−1^) solution was used for back titrations of the filtrate to determine the oxygenated groups concentration. Similarly, NaOH (0.01 M) standard solution was used for back titration to estimate basic groups contents in the filtrate after stirring the 0.15 g activated carbon in 50 mL of 0.01 M HCl for one day.

### 2.4. Characterization of mAC and Fabricated Membranes

The membrane samples were cut into the standard sizes and dried thoroughly under vacuum before analysis. The functional groups present on mAC and the fabricated membranes surface were investigated using ATR-FTIR Bruker ALPHA spectrophotometer. Scanning electron microscopy (SEM) (JEOL JSM 6490A) was operated to take the micrographs of the AC, mAC, surface and cross section of membranes. To analyze cross section, membranes were immersed into liquid nitrogen to give a clean crack. Cracked samples were dried and then mounted on steel studs for sputter coating before analysis. Surface area of mAC and fabricated membranes were determined by Brunauer-Emmett-Teller (BET) (Micrometrics) nitrogen adsorption/desorption technique. Samples had been degassed at 120 °C under high vacuum for 6 h and relative pressure range was between 0.05–0.09. The membrane surface roughness was determined by NANOVEA PS-50 (optical profilometer). The membrane samples were placed on the platform and downward scan of 150 μm bipolar was used for surface roughness (Ra). UTM AG-XD plus was used to estimate the tensile strength and elongation percent at a loading velocity of 100 mm/min. In order to analyze wetting properties, the surface hydrophilicity/hydrophilicity of prepared membranes was measured using DSA 25, KRUSS contact angle instrument. The contact angle was estimated at five random positions for each sample by using de-ionized water and the average time was calculated in order to eliminate the experimental error. A static sessile drop method was used in which 10 μL deionized water was dropped onto the membrane surface.

For measuring apparent free energy of the surface Chibowski approch, young equation, and equilibrium contact angle were applied as in Equation (1) [37,38].
(1)$$\gamma_{s}=\frac{\gamma_{l}}{2}\left(1+cos\theta_{eq}\right)$$
where $\theta_{eq}$ is the equilibrium contact angle, $\gamma_{l}$ is the liquid surface tension, and $\gamma_{s}$ is the apparent surface free energy.

Water uptake (*WU*) is the ability of a membrane to absorb water. The membrane samples were cut into standard sizes and drenched into the ultrapure water for 24 h. The initial mass of the membranes was calculated when they were wet i.e., $W_{wet}$. The wet samples were than dried at 60 °C for 12 h and weight again in dry state i.e., $W_{dry}$. The values were calculated three times in order to reduce experimental error. WU (%) was calculated using Equation (2) [39].
(2)$$WU\;\left(\%\right)=\frac{W_{wet}-W_{dry}}{W_{dry}}\times 100$$

The power of developing membranes upon interaction with water is known as swelling of membrane [40]. It is closely related to *WU* ability of a membrane. The test was conducted in same manner as the *WU* but instead of weight, the difference in lengths (*L*) of each membrane was calculated. Swelling (%) was calculated by using Equation (3) [41].
(3)$$Swelling\;\left(\%\right)=\frac{L_{wet}-L_{dry}}{L_{dry}}\times 100$$

Flux can be defined as fluid volume passed through certain membrane per unit area per unit time. A custom-made dead end filtration assembly having filtration area 0.002 m^2^ was used to perform the test under the pressure of 0.1 MPa at constant stirring rate of 100 rpm. Schematic of the filtration experiment is presented in Figure 2.

The membrane permeation flux (J) for water was estimated through Equation (4) [42]:(4)$$J=\frac{V}{A\times t\;}$$
where V is the volume (L) of water, A is the effective membrane area (m^2^) and t is the time (h).

BSA solution 1000 ppm with ionic strength 0.1 was filtered through the fabricated membranes at 0.1 MPa pressure and the concentration in permeate was obtained from the absorbance at 278 nm using a calibration curve in accordance to the Beer-Lambert’s law. For measuring BSA concentration at 280 nm, UV-spectrophotometer (SHIMADZU, Kyoto, Japan) was used for the BSA solution. The rejection of BSA was calculated at IEP 4.7 at pH 8.22 and 0.1 M KCL solution by using Equation (5).
(5)$$R\left(\%\right)=\left(1-\frac{C_{p}}{C_{f}}\right)\times 100$$
where $C_{p}$ and $C_{f}\;$are permeate and feed concentrations respectively (mg L^−1^).

### 2.5. Antibacterial Assay

Antibacterial activity of the fabricated membranes against gram-negative *E. coli* (ATCC 15224) and gram-positive *S. aureus* (ATCC 6538) was evaluated by disk diffusion method. Before performing antibacterial activity bacterial cultures were activated by speckling the bacteria on freshly prepared MHA and these plates were put into oven at 37 °C for 24 h. A single colony form newly grown bacterium was picked and optical density (OD) was set to 0.5 by McFarland standard. Microbial broth was vortex for 5 s to get uniform dispersion. The sterilized MHA agar was poured into sterile petri dishes after solidification of agar 100 µL from prepared culture media was lined evenly on the agar plate by sterile cotton swab. Membrane samples were cut into 6 mm in diameter disks by 6 mm cylindrical punch, positioned on a petri plates and incubated at 37 °C for 24 h. Zone of inhibition were measured using digital caliper around each membrane sample. All experiments were performed at least three times, and the results are presented as the mean ± standard deviation and t test was used to determine the statistical significance (* p < 0.05, ** p < 0.01, *** p < 0.001, **** p < 0.0001) [43]. Ciprofloxacin/DMSO were used as positive/negative controls. dia of membrane disk (d) and inhibition zone ($d_{iz}$) have been measured to calculate the normalized width of the antimicrobial “halo” (*nw_halo_*) [44] of each disk by using Equation (6).
(6)$$nw_{halo}=\frac{\frac{d_{iz}-d}{2}}{d}$$

## 3. Results

### 3.1. Characterization of AC

#### 3.1.1. Surface Chemistry

For AC the pH of the null charge point shows the neutral value, (pH_PZC_ 7.4 for HCl treated granulated AC). Boehm titration results, pH_PZC_, and ash contents of AC was summarized in Table 2. As stated in Table 2, the AC contains very small concentration of oxygen containing surface groups, composed mostly of carbonyl groups (0.48 meq.g^−1^) whereas the number of phenolic groups found to be higher (~0.2 meq.g^−1^). The acid-treated AC (0.1%) has a relatively little ash content relative to a pristine substance (about 9%), suggesting that mainly the metal impurities have been eliminated considerably by washing.

#### 3.1.2. Surface Characteristics Analyses by Brunauer-Emmett-Teller Analysis (BET)

Type I nitrogen adsorption-desorption isotherms for AC were seen in Figure 3, which is characteristic of the microporous texture. The measured mesoporous volumes of 0.07 cm^3^ g^−1^ were negligible as outlined in Table 3. It is evident that Type I isotherms with a given knee and long platform extending to P/P_o_ to 1.0 suggest that the AC is basically microporous solid, as seen by a substantial increase in adsorption with low P/P_o_ (<0.1). A lack of hysteresis revealed that mesoporosity was deficient, and mostly micropores are present in the carbon materials with just a limited proportion of mesopore. [45].

In a relative pressure range (0.01 to 0.05) the BET specific surface area (SBET) of AC was calculated to avoid negative unrealistic C factor. SBET is here an indicator of the volume of the micropores rather than a genuine estimation of the surface. The S_BET_ of AC found to be 1044 m^2^ g^−1^ and the total microporous volume 0.53 cm^3^ g^−1^) [46].

The pore size distributions (PSDs) of ACs determined by DFT also prove to be microporous in nature. AC has few larger pores and therefore suggested the structure of small mesopore as shown in Figure 4. PSDs for pores below 3 nm in Figure 3 presents the incremental pores below 10^−3^ cm^3^ g^−1^ that are insignificantly estimated within the range (2.3–25 nm) indicating granulated AC contains various types of micropores [47]. More than 50% of the volume is ultra-micropores (pore size < 0.8 nm) represented in Table 3. In comparison with micropores as shown in Figure 4 and Table 3, the mesoporous quantities of DFT-determined activated carbons (pores with a diameter of more than 2 nm) were insignificant (0.07 cm^3^ g^−1^).

#### 3.1.3. Surface Morphology Analyses by SEM

The AC morphology and pore size were observed by Scanning electron microscopy (SEM) analysis. Images of unmodified AC at 875X magnification are shown in Figure 5a,b. The surface tends to be smooth and tiny pores are scattered around it. The remaining salt or other compounds on the AC, certain particles are dispersed on the AC surface [48]. Figure 5c,d shows the SEM images of functionalized activated carbon. The images indicated the porous structure of AC after it has been functionalized. The functionalized AC appears to be in the form of large aggregates of smooth surface particles without visible porosity. The BET surface area and pore volume also decreases for mAC. This can be explained on the basis of diffusion rate and size of functionalization reactants in to the pores of AC. The NH_2_ functional group has smallest size and better diffusion rate into the pores of AC, which results in reduction of surface area and pore volume [49].

#### 3.1.4. Afunctional Groups Analyses by FTIR

ATR-FTIR spectra of pristine activated carbon and APTMS modified activated carbon is shown in the Figure 6. In the spectra the, the weak and broad peak at 3391 cm^−1^ indicates the amine stretching and the peak at 1540 cm^−1^ indicates the N-H bending [50]. These two peaks represent the amine functional group due to the functionalization of AC with APTMS. The spectra clearly indicate the functionalization of AC by APTMS.

### 3.2. Characterization of mAC and Fabricated Membranes

#### 3.2.1. Functional Groups Analyses by FTIR

Figure 7 shows the ATR-FTIR spectra of the APTMS modified mAC with amino groups on the surface of various fabricated membranes. The amine stretching at 3485 cm^−1^ and 1645 cm^−1^ conforming the functionalization of AC by APTMS [51]. MP0 indicates the pristine PES membrane. All membranes had usual peaks of PES, such as the 1585 cm^−1^ benzene ring stretching, the 1495 cm^−1^ C=C bond, the 1247 cm^−1^ and 1170 cm^−1^ aromatic sulfone group [52,53]. In the spectra the broad peak at 3450 cm^−1^ indicates the presences of hydroxyl group that is due to the presences of PVP used as pore former during the fabrication of PES membrane. The peak of 1637 cm^−1^ indicates the amine group of PVP. AC-containing membrane was allocated for the hydroxyl groups of some functional groups including carboxy, carbonyls and phenols to the wide absorption range of 3250 cm^−1^ and 3430 cm^−1^ [54,55].

#### 3.2.2. Surface Morphological Analysis by SEM

Phase inversion via immersion precipitation method was used for the fabrication of flat sheet membranes and the membrane surface seems to be homogenous as visualized in Figure 8. The fabricated pristine membrane and membranes prepared with a variable volume of AC powdered as a solvent in DMAc are shown in Figure 8. The compositions of casting solutions for the prepared membranes are summarized in Table 1. To further analyze the surface and morphology of the prepared membranes, SEM and profilometry analyses were carried out.

Figure 9 and Figure 10 present the SEM images of the fabricated membranes. Surface pores with non-uniform distributions were found for the pristine membrane and several ultra-fine failures and cracks were noticed. However, for membranes incorporated the activated carbon, their addition altered the morphological structure with observation of the sponge shape [55]. Hydrophilicity is the reason for the disparity between the two porous membranes. It was considered that solvent and non-solvent exchange rate is influenced by porosity and adsorbent hydrophilicity during the inversion stage. By increasing the diffusion rate of solvent from the cast film to the water bath helps in the formation of fingerlike voids were created [56]. The high adsorption potential of AC induces solvent adsorption and improved casting solution viscosity by means of effective physical interactions. The casting solution with AC improved in viscosity to the extent where the diffusion rate of the solvent was decreased [57]. This led to a slightly porous structure and a morphology similar to the sponge with uniform smaller voids. Adding materials modified the structure of the membrane support layer and increased the hydrophilicity of surface [58].

#### 3.2.3. Optical Profilometry

Various characteristics of the fabricated membranes are summarized in Table 4 including surface roughness. Membrane surface roughness is highly efficacious parameter in its various features regarding fouling parameters, therefore prepared membranes specimen surface roughness was investigated by optical profilometry and results are presented in Table 4. MP0 shows less surface roughness due to the presences of PVP that results in the improving the surface coverage of PES membrane [59]. MC1, MC4, MC7 and MC10 shows an increase in the surface roughness than MP0 that is due to the incorporation of activated carbon into PES membrane. These results are possibly due to the meso and microstructures of activated carbon that contribute to an increase in membrane porosity [25]. Thus, the roughness in the PES membrane region increases by increasing the percentage of activated carbon. For the mAC embedded PES membrane, a “nodular” topology and interconnected cavity channels were found [55].

#### 3.2.4. Surface Characteristics Analyses by Brunauer-Emmett-Teller Analysis (BET)

Concerning to the modification of AC with APTMS, a decrease in BET area and pore volume of mAC up to 188 m^2^/g and 0.490570 cm^3^/g respectively, in relation to non- chemically modified AC. The formation of new surface functional groups attributed as the major obstruction for the drastic decrease in surface area [60]. Similar results have shown that formation of surface functional groups like nitriles, amides, amines, among others block the existing pores and reduce the surface are upon chemical modification of carbonaceous material [61,62].

The BET area and pore volume of fabricated membranes are shown in Table 4. The surface area increased from 0.5173 to 1.1961 m^2^/g for pristine and modified membranes respectively. This can be attributed due to the presence and dispersion of mAC particles with high surface area in polymer matrix. Figure 7 also depicts the suitable dispersion of mAC particles. The contact area to pass the water through surface of particles and membrane is increased by uniform distribution of mAC powder in membrane structure. Interestingly, MC10 has less surface area value as compared to other modified membranes. This is due to the agglomeration of mAC powder in membrane structure, which can also depict from Figure 9, showing sponge like structure. Agglomeration of the powder leads to pore blockage and thus decreasing the existing surface area of the membrane structure [63]. Studies shows that functionalization of AC aims to prevent the agglomeration but beyond certain loadings agglomerates still form and pore blockage by mAC became more prominent [64].

#### 3.2.5. Contact Angle and Surface Energy Studies

The contact angle of the pristine membrane and membranes incorporated with mAC are presented in Figure 11 and properties are summarized in Table 4. MP0 shows the contact angle of 69° which is slightly lesser than pure PES as addition of PVP increases the hydrophilicity due to presence of hydroxyl groups in its structure [65]. The composite membranes show small contact angle i.e., up to 51° in comparison with MP0. This lowering in the contact angle can be attributed for incorporating higher concentration of mAC in PES membranes [66]. As AC have meso- and micro- structure and having various polar presents in the structure that helps in decrease in contact angle, thus by increasing the hydrophilic character of the membrane. MC10 shows contact angle of 58°, this slight increase in contact angle of MC10 is possible due to the agglomeration of AC beyond certain concentration [25,67]. Based on the equilibrium water contact angle, the apparent surface free energy was measured. With decreases in balance water contact angles as reflected in Figure 11, surface energy is increased. The approximate surface-free energy values are based on the physical and chemical characteristics of water.

#### 3.2.6. Water Uptake and Swelling

For water treatment applications *WU* and swelling are important parameters for membranes to investigate, as they specify the filtration performance and productivity. These two properties are strongly related to the presence of hydrophilic components and functional groups on the membrane surface [68]. Figure 12 shows the *WU* and swelling percent of fabricated membranes. The pristine membrane exhibits very less *WU* and swelling % indicating the hydrophobic nature of PES. Which is in agreement with the contact angle and SEM results; indicating high contact angle and thin asymmetric top layer of PES membrane results in less *WU* and swelling ability. The modified membranes show better results as compared to the pristine due to the addition of hydrophilic mAC. Because of the hydrophilicity and highly porous structure of mAC it has positive impact on *WU* and swelling % of membranes [1,69]. The % of water uptake and swelling increases as amount of mAC is increased up to 7% (2-times as compared to pristine) relative to its dry state beyond this value the agglomeration of mAC particles dominates and respective % value starts decreasing. Pore volume also plays a critical role in water absorption capacity of membranes, as pore volume increases the water uptake and swelling percent also increases [70]. As MC7 has highest surface are and pore volume thus, it has more ability to absorb water molecules and better antifouling properties. The values of *WU* and swelling (%) are given in Table 4.

#### 3.2.7. Evaluation of Flux and Fouling Characteristics

Figure 13 represents the flux, fouling and rejection characteristics of the prepared membranes. As the concentration of mAC in the membranes increased, the permeating water flux was observed to increase up to mAC concentration of 7% relative to pristine MP0 membrane. However, at higher mAC concentrations i.e., beyond 7%, water flux declined as presented in Figure 13a. This intriguing behavior can be attributed to the limitation in the network distribution of very fine activated carbon particulate inside the polymer matrix. This can influence the pore density and functionality to effect water flux. There are studies that observed increase in water flux for the relatively lower AC loading in to the polymer membrane [71]. In present work, much higher loading of activated carbon was explored that can influence the surface area, roughness and contact angle in such a way to produce better flux. There is a possibility of optimize mAC concentration to produce enhancement in flux in polymer membranes. Due to more accumulation, weak dispersion at higher concentrations of mAC and formation of disconnected channels results in lower water flow [72].

The BSA protein rejection behavior of fabricated membranes is presented in Figure 13b and showed that the membrane incorporated mAC have significantly lower BSA rejection. The MC4 and MC7 membrane enabled to permeate over 80% of the protein to exhibit cleanliness non-binding adsorption and antifouling behavior shown in Figure 13a. Contrary to this, pristine MP0 membranes fouled respectively with over 70% of the BSA protein. The fabricated membranes with higher loadings of the mAC provided excellent filtration permeation for the BSA proteins at its higher concentration of 1000 ppm. Compared to pristine membrane MP0, MC types of membranes enable to provide significantly lower binding of proteins molecules to its surface to prevent loss of analytes during the process of filtration. Furthermore, these MC membranes were also observed to be cleansed as compare to their filtration performance against filters employed in syringes for biological samples separation. This characteristics of MC membrane provide advantages since it did not induce chemical contamination during potential applications of many filtration processes of biological samples. Thus, these membranes can be the best option for the filtration of biological samples. Others PES pristine membrane MP0 caused BSA analyte to adsorbed and fouled as ‘sticky’ proteins.

The interesting observation of relatively higher permeation and lower rejection of the BSA protein from the mAC incorporated membranes can be attributed to combination of several factors. The filtration and rejection characteristics of proteins on carbon are complex and dependent on its surface characteristics, the isoelectric point (IEP) of the protein, and pH of the solution [73]. Since, filtration of the BSA protein solution was carried at the IEP value of 4.7 at pH 8.22 in water at 25 °C, therefore, its surface is expected to be more hydrophobic and compact. Therefore, the rejection at IEP is expected to be very low as compared to other pH valuesl [74]. It is known that BSA molecules can fold and molecular hydrophobic portions expose to the membrane embedded with AC surface lead to pass from the surface. This possibility is supported from the higher flux as well as a decrease in BSA rejection. Further evaluation of activated carbon to separate proteins would be of interest to optimize the purification of protein mixtures of higher complexity.

The highest flux for PES UF membranes has been shown by mAC filler added membranes which is in agreement with reported results [27]. The highly porous structure, large surface area and hydrophilic character of mAC makes the membranes more suitable candidate for UF process. It has been reported that PES UF membranes incorporated with charged AC particles are more stable towards fouling resistance and hence better permeability [26]. The composite membranes have better properties in terms of porosity, surface area, hydrophilicity, fouling resistance, permeability and long-term stability. Thus, AC incorporated UF membranes are potential candidate for water treatment applications. The ultrafiltration mechanism of reported membranes is summarized in Table 5.

#### 3.2.8. Antibacterial Activity of the Fabricated Membranes

The fabricated membranes were studied for their bacterial resistance against *S. Aureus* and *E. Coil,* and observation are presented in Figure 14 and Figure 15. Figure 14 illustrates the antimicrobial “halo obtained from the measurements of the normalized width, inhibition zone diameter ($d_{iz}$) and the diameter of disk (*d*). As shown in Figure 14, after incubation for 24 h at 37 °C, the results of the agar disk diffusion method clearly indicate no antibacterial activity for pristine membrane. However, increasing antibacterial activity were observed against *S. aureu* and *E. coil* for the mAC membranes. Figure 15 shows the different normalized widths of the antimicrobial “halo” (*nw_halo_*) for the fabricated membranes against gram-positive *S. Aureus* and gram-negative *E. coli* bacteria. The differences are statistically significant (*p* < 0.0001).

The fabricated membranes show bacterial resistance due to the presence of mAC in the prepared membranes. It was observed that higher concentration of mAC produced more pronounce antibacterial effect. The results showed better zone of inhibition for the mAC incorporated PES membranes, thus making these well-suited filtration for treatment of water. Present observation is in good agreement with earlier studies of antibacterial properties of activated carbon against various bacteria including *E. coli* [73,78,79,80,81].

The membranes developed and studied in present work for water treatment possess promising antimicrobial properties to prevent biofouling since later leads to pores clogging to render these unfit for filtration operation. The antimicrobial performance of the prepared PES ultrafiltration membranes with mAC thus can potentially be utilized to suppress antifouling to significantly enhance the performance and longevity of filtration through membranes.

## 4. Conclusions

In this work, high performance ultrafiltration polyether sulfone membranes incorporated with the APTMS modified activated carbon (mAC) have been fabricated via inversion phase process by dispersing mAC into the coagulation bath. Influences of a range of concentrations of mAC with 1, 4, 7 and 10% (*w/w*) on the fabricated membranes was investigated such as morphology, hydrophilicity, flux, protein and bacterial antifouling. The incorporation of mAC into the membrane resulted in hydrophilicity as contact angle decreased from 73° for pristine PES membrane to 55° for membrane incorporated with 7% (*w/w*) mAC with increase in water flux up to 136 (L/m^2^·h). The water flux also enhanced notable relative to the pristine PES membrane in all the mAC membranes. The prepared membranes incorporated with mAC also exhibited higher antibacterial performance against both strains of gram-negative *E. Coil* and gram-positive *S. aureus*. The bovine serum albumin protein (BSA) solution filtration through mAC membranes of 4% (*w/w*) and 7% (*w/w*) resulted in over 90% as compared to only 30% for the pristine PES membrane to demonstrate antifouling characteristics. The novel mAC PES ultrafiltration membranes have higher antifouling and non-binding of the protein and bacteria and therefore can potentially be useful for water treatment and selective filtration of proteins and macromolecules.

## Figures and Tables

**Figure 1 polymers-14-02264-f001:**
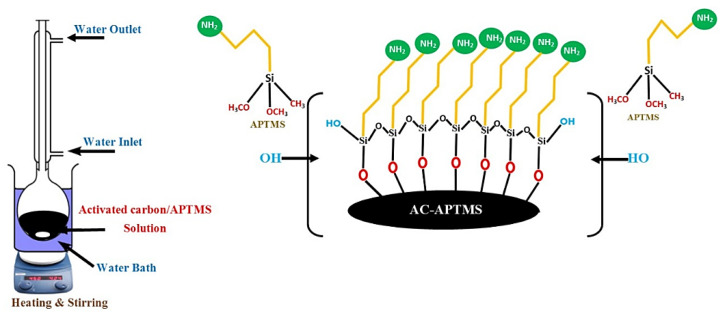
Schematic of amine modified activated carbon by reflux.

**Figure 2 polymers-14-02264-f002:**
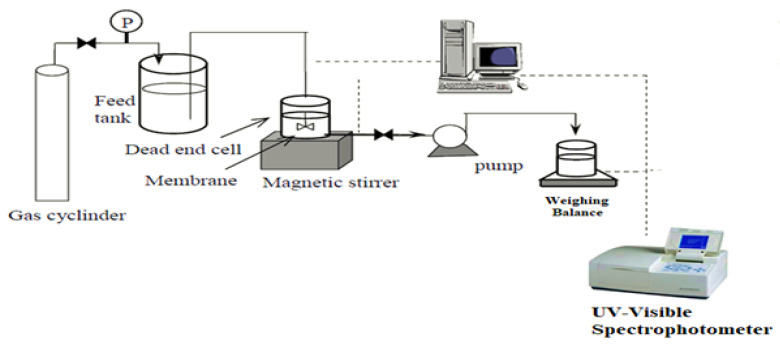
Dead end filtration assembly to check permeation flux and absorbance.

**Figure 3 polymers-14-02264-f003:**
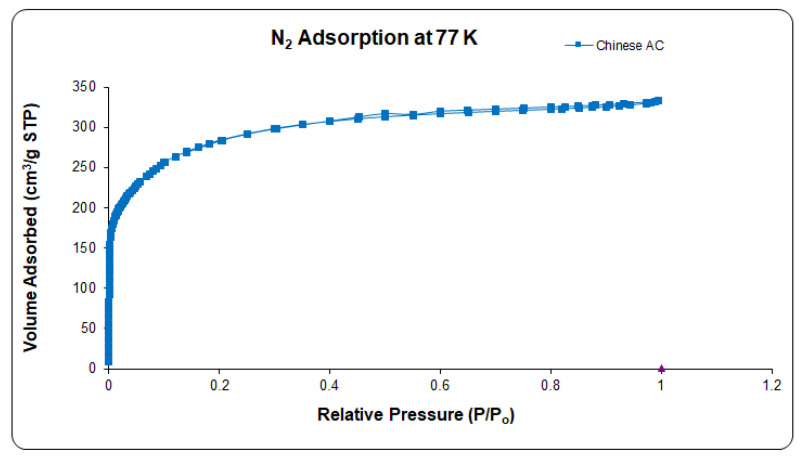
Adsorption-desorption isotherms of N_2_ at 77 K for the AC.

**Figure 4 polymers-14-02264-f004:**
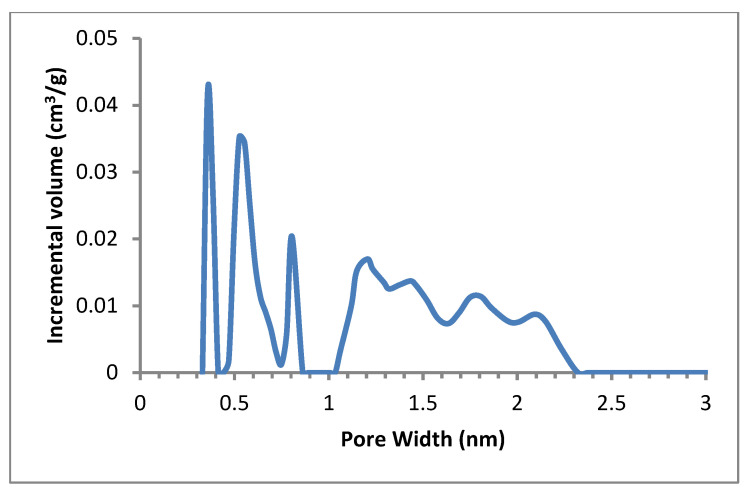
PSD from N_2_ adsorption isotherms at 77 K. The PSD was acquired using method of 2D-NLDFT.

**Figure 5 polymers-14-02264-f005:**
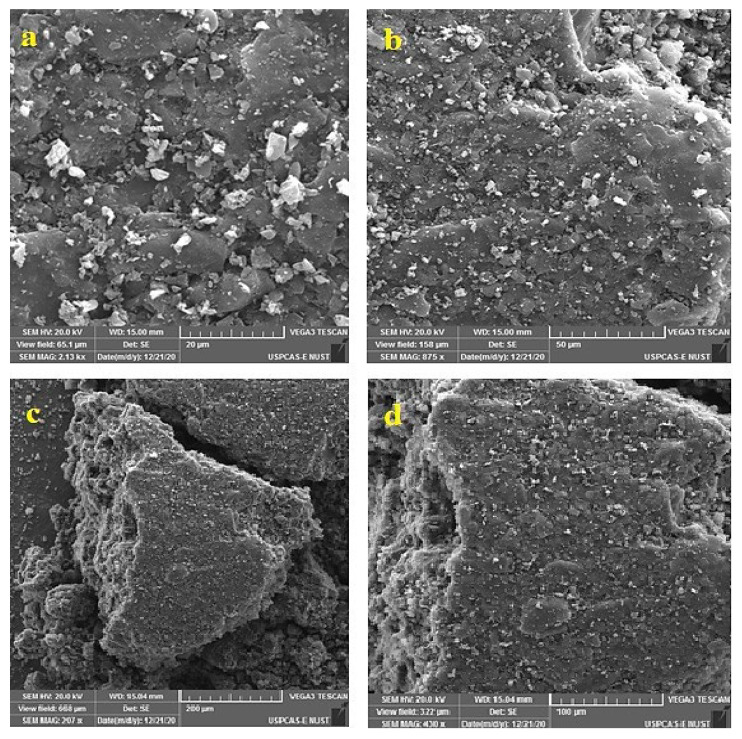
SEM micrographs of (**a**,**b**) Pure AC and (**c**,**d**) mAC.

**Figure 6 polymers-14-02264-f006:**
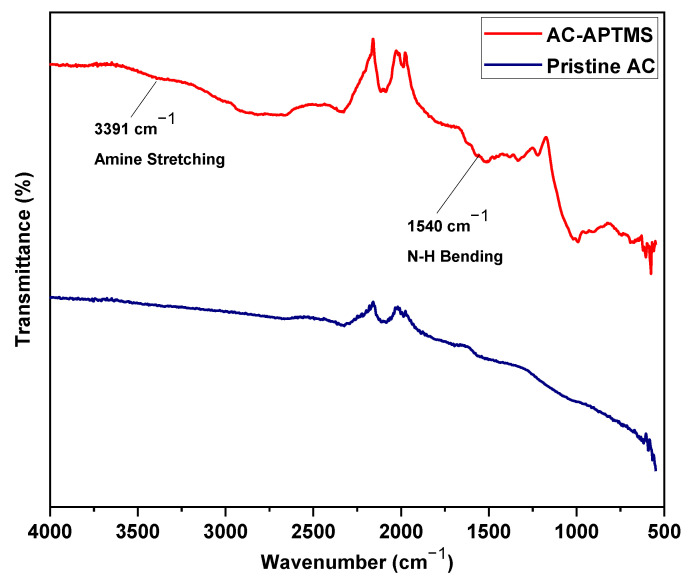
ATR-FTIR of pristine and APTMS functionalized AC.

**Figure 7 polymers-14-02264-f007:**
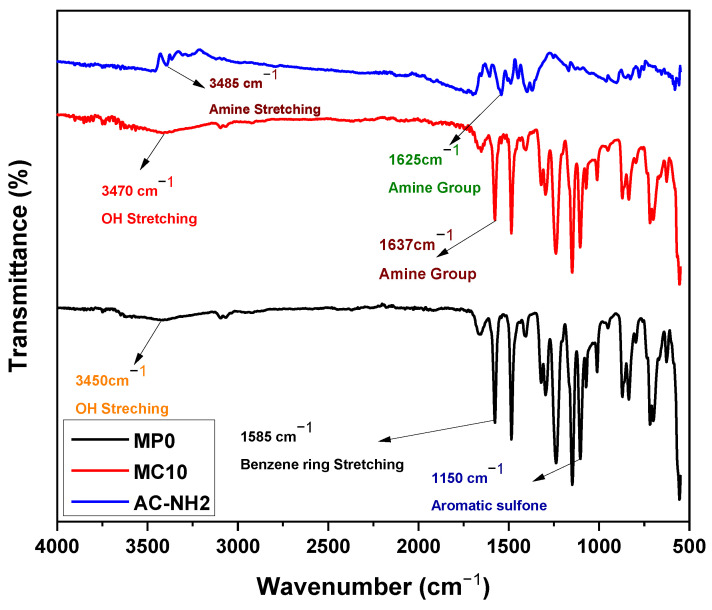
FTIR of MP0 Pristine PES, MP10 PES with AC, Modified AC.

**Figure 8 polymers-14-02264-f008:**
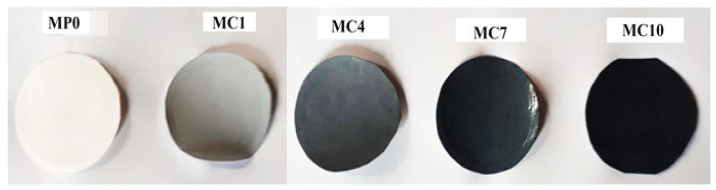
Fabricated pristine PES membrane and mAC incorporated PES membranes. Sample codes are summarized in Table 1.

**Figure 9 polymers-14-02264-f009:**
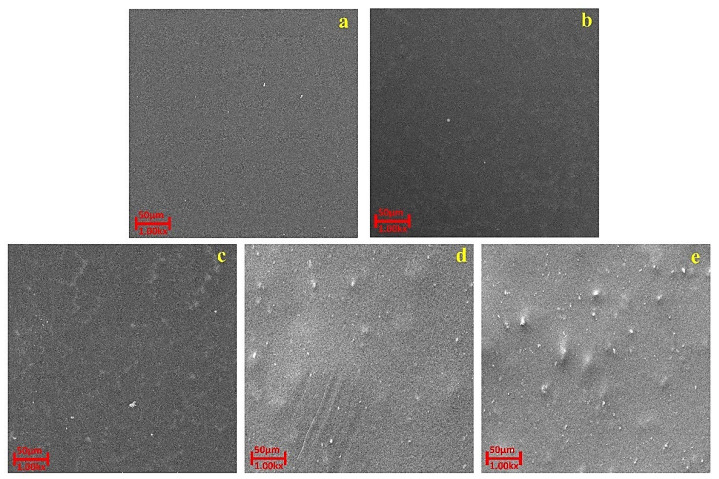
Surface topography SEM images of (**a**) MP0 (**b**) MC1 (**c**) MC4 (**d**) MC7 (**e**) MC10. Sample codes are summarized in Table 1.

**Figure 10 polymers-14-02264-f010:**
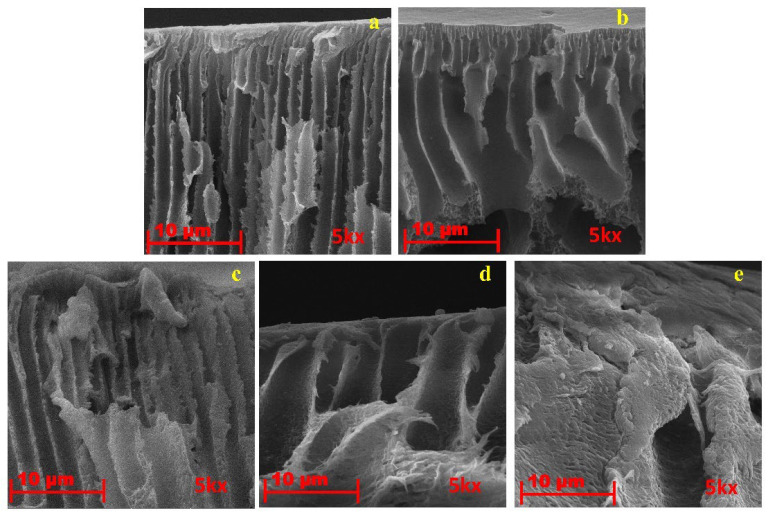
Cross-section SEM images of (**a**) MP0 (**b**) MC1 (**c**) MC4 (**d**) MC7 (**e**) MC10. Sample codes are summarized in Table 1.

**Figure 11 polymers-14-02264-f011:**
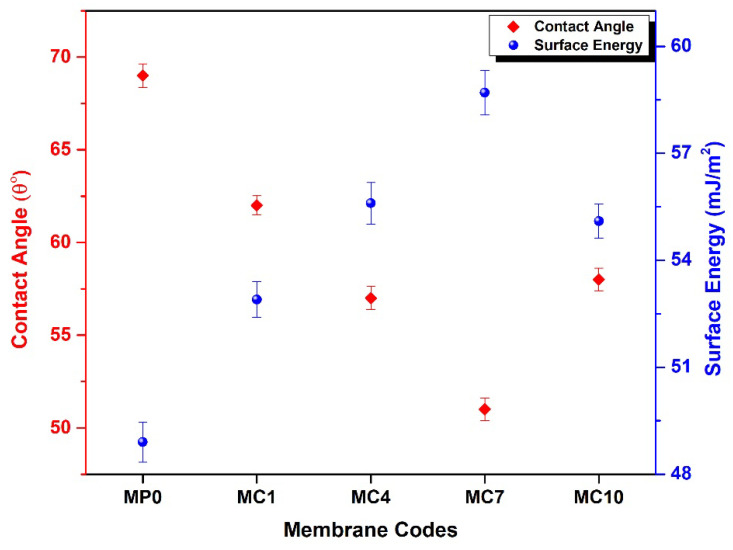
The contact angle of pristine PES membrane and mAC incorporated PES membranes. Sample codes are summarized in Table 1.

**Figure 12 polymers-14-02264-f012:**
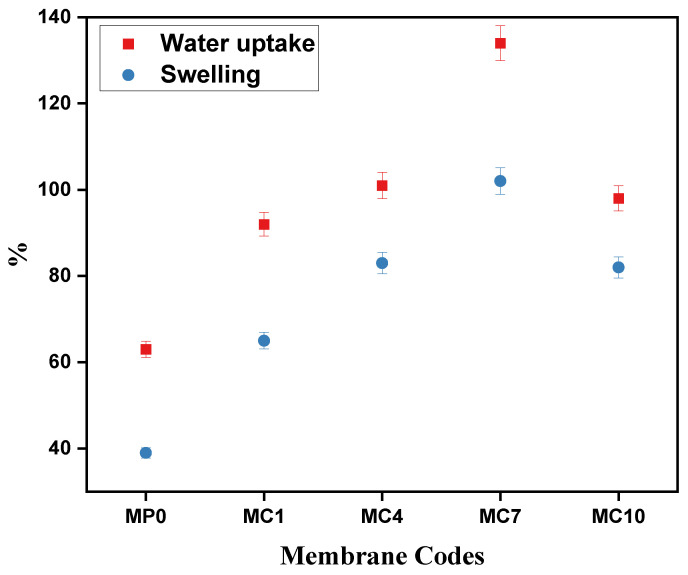
The *WU* and Swelling % of pristine PES membrane and mAC incorporated PES membranes. Sample codes are summarized in Table 1.

**Figure 13 polymers-14-02264-f013:**
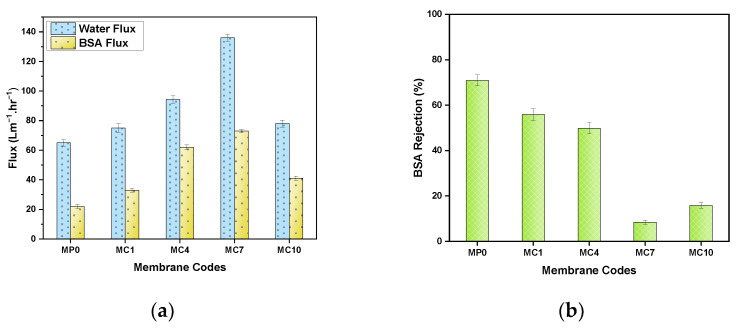
Studies of the performance of the fabricated membranes of the pristine and mAC membranes: (**a**) Water and BSA flux rates; (**b**) BSA rejection (%).

**Figure 14 polymers-14-02264-f014:**
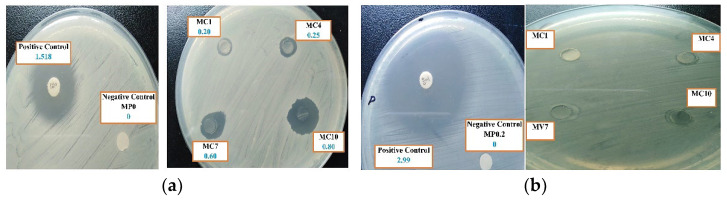
Images of surfaces of agar diffusion studies of the fabricated membranes to (**a**) *S. aureus* and (**b**) *E. coli* inoculums, respectively.

**Figure 15 polymers-14-02264-f015:**
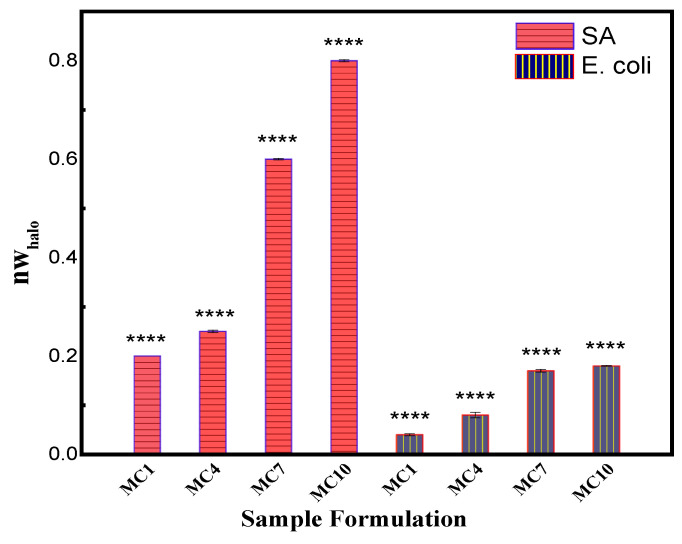
The normalized width of the antimicrobial halo for the gram-negative *E. coli and* gram-positive *S. aureus* bacteria. (**** p < 0.0001).

**Table 1 polymers-14-02264-t001:** Membrane fabrication using 18% (*w/w*) of PES, PVP and by varying DMAc solvent, modified activated carbon.

Code	PVP	DMAc	mAC
MP0	0.2%	81.8%	0%
MC1	0.2%	80.8%	1%
MC4	0.2%	77.8%	4%
MC7	0.2%	74.8%	7%
MC10	0.2%	71.8%	10%

**Table 2 polymers-14-02264-t002:** Results of Boehm titration, pH_PZC_ and ash contents of AC.

Carboxylic Groups (meq.g^−1^)	Lactonic Groups (meq.g^−1^)	Phenolic Groups (meq.g^−1^)	Carbonyl Groups (meq.g^−1^)	Total Basic Groups (meq.g^−1^)	pH_PZC_	Ash Content (%)
0.00	0.20	0.20	0.48	0.53	9.5	5

**Table 3 polymers-14-02264-t003:** Textural properties of AC by N_2_ adsorption at 77 K and CO_2_ adsorption at 273 K.

BET Surface Area S_BET_ (m^2^·g^−1^)	Ultramicropore Volume * (Ø < 8 nm) (cm^3^·g^−1^)	Supermicropore Volume ^#^ (Ø > 8 nm) (cm^3^·g^−1^)	Micropore Volume (cm^3^·g^−1^)	Mesoporous Volume (cm^3^·g^−1^)	Total Pore Volume ^$^ (cm^3^·g^−1^)
1044	0.26	0.21	0.47	0.07	0.53

* obtained from DFT slit pore model applied on CO_2_ adsorption data at 273 K. ^#^ obtained from DFT slit pore model applied on N_2_ adsorption data at 77 K. ^$^ obtained from single point adsorption.

**Table 4 polymers-14-02264-t004:** Characteristics of the fabricated PES pristine and modified AC incorporated membranes.

Properties	MP0	MC1	MC4	MC7	MC10
Contact Angle (θ)	69	62	57	51	58
Water uptake (%)	63	92	101	134	98
Swelling (%)	39	65	83	102	82
Roughness Ra (µm)	0.212	0.423	0.619	0.814	0.981
BET surface area (m^2^/g)	0.5173	0.5376	1.0041	1.7126	1.1961
Pore volume (cm^3^/g)	0.001539	0.001718	0.003058	0.004715	0.003432
Water flux (L/m^2^·h)	65	75	94	136	78
BSA flux (L/m^2^·h)	22	33	62	73	41
BSA rejection (%)	70.9	55.9	49.8	8.4	15.8

**Table 5 polymers-14-02264-t005:** Ultrafiltration mechanism of membranes reported in literature.

Membrane Material	Method	Contact Angle	Pure Water Flux (L/m^2^·h)	Pressure (MPAa)	Fouling Characteristics	Reference
PES, APTMS modified AC	Phase inversion	51	136	0.1	BSA rejection 8.4% Zones of inhibition for *S. aureus* 4.8 mm *E. coli* 2.99	Our work
PES incorporated Functionalized AC	Phase inversion	106	33	0.1	Reduction in COD, BOD and TDS level	[66]
PES-C/emodin ultrafiltration membrane	Phase inversion	-	350	0.2	*S. aureus* 3 mm Zone of inhibition	[75]
carboxylic acid functionalized polysulfone, PES, PVP	Phase inversion	74	400	0.2	BSA rejection 4% wt	[76]
PES, Acacia Gum	Phase inversion	65	70	4	Antibacterial against *E. coli*, Low BSA rejection	[77]

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
