# Peer review of "Water Treatment Using High Performance Antifouling Ultrafiltration Polyether Sulfone Membranes Incorporated with Activated Carbon"

_polymers, 2022, doi:10.3390/polym14112264_

Round 1

Reviewer 1 Report

In this manuscript, PES ultrafiltration membranes with enhanced antifouling properties were fabricated via the phase-inversion method by incorporating varying concentrations of AC modified with APTMS. The higher flux and antifouling characteristics of the mAC-modified membranes were attributed to the electrostatic repulsion from the unique functional properties of AC and network structure of APTMS. The novel composite ultrafiltration membranes can provide higher flux and less BSA rejection and thus can find antifouling applications for the isolation and concentration of proteins and macromolecules.

I consider the content of this manuscript will definitely meet the reading interests of the readers of the Polymers journal. Therefore, I suggest giving a minor revision and the authors need to clarify some issues or supply some more data to enrich the content.

  • The length of the abstract obviously exceeds the requirements of the journal, which needs to be appropriately reduced and the language needs to be refined. ‘Abstract: The abstract should be a total of about 200 words maximum.’ (https://www.mdpi.com/journal/polymers/instructions)

  • For the Keywords, ‘ultrafiltration membrane’, ‘phase-inversion method’, and ‘antibacterial performance’ should also be added to attract a broader readership and highlight the significance of this work.

  • Please pay attention to grammar and spelling problems, especially the missing or redundant definite articles. I suggest double-checking the whole manuscript. I will point out several examples, but unfortunately, I cannot point out all of them. For example:

Line 45, ‘The availability of clean water is essential for the human population and vital for the sustainability of developing countries’;

Line 118, ‘This dried AC was referred to as mAC consisting of amine functional groups on the surface’;

Line 129, ‘PVP and mAC were the subsequently added to the PES solution and continuously stirred for the next 24 h’;

Line 379, ‘Interestingly, MC10 have surface area has less surface area value as compared to other modified membranes’ and so on.

  • Line 48, ‘To date, various carbon-polymer composite materials like graphite, carbon nanotubes, graphene carbon mesosphere and AC are well known for different water treatment applications’. Indeed, none of the listed materials really belong to carbon-polymer composite materials, since no polymers are listed. They are just carbon materials, so they should be ‘various carbon materials like graphite, carbon nanotubes...’

  • Line 112, ‘2.2.1. Surface Modification of Activated Carbon (AC)’, for the APTMS-activated carbon, why is only one ratio used between APTMS and AC? Is there any specific reason for this selection of ratio? Or I suggest supplying more ratios for the optimization of the best loading of APTMS into AC.

  • Line 194 to 196, ‘Surface area of mAC and fabricated membranes were determined by Brunauer-Emmett-Teller (BET) (Micrometrics) nitrogen adsorption/desorption technique.The sentence should be deleted, since in ‘2.3.2. N2 Adsorption-Desorption’, has already been described once.

  • Except for contact angle measurement, water uptake of the membrane should also be supplied as a physicochemical property and may also demonstrate the hydrophilicity of the obtained membranes. See section 2.5 of [Electrochimica Acta 309 (2019): 311-325].

  • Line 293, ‘Figure 5 (c,d) shows the SEM images of functionalized activated carbon. The images indicated the porous structure of AC after it has been functionalized.’That is true, but this can be simply understood from the caption of Figure 5. So before and after the functionalization of AC, how does the porous structure change exactly? What is the conclusion that readers should draw from the SEM? This needs to be clarified and explained better.

  • Line 311, ‘such as the 1585 cm-1 benzene ring stretching, the 1,495 cm–1 , C–C bond, the 1,247 cm–1 aromatic ether ring and 1,170 cm–1 aromatic sulfone band. [46].Here, 1495 should also be aromatic ringνs [C=C], 1247 should be νs [O=S=O]. The authors should double-check the peak assignments [Electrochimica Acta 309 (2019): 311-325; Journal of colloid and interface science 300.1 (2006): 286-292].

  • Line 425, ‘The fabricated membranes with higher loadings of the PAC provided excellent filtration permeation for the BSA proteins at its higher concentration of 1000 ppm.What is PAC? It has never been used in the text before, and I consider it should be mAC.

  • Line 457, ‘As shown in Figure 14, after incubation for 24 h at 37 °C, the results of the agar disk diffusion method clearly indicate no antibacterial activity for pristine membranes MP0 and MP0.2.’It is clear that the pristine membrane is MP0. And what is MP 0.2? It never appears previously, and how can MP0.2 belong to a pristine membrane as MP0? I consider it may be the sample with a tiny amount of mAC. But it needs to be explained in detail. Previously, it only refers to ‘concentrations of mAC with 1, 4, 7 and 10% (w/w) on the fabricated membranes’, nothing related to 0.2.

Reviewer 2 Report

This manuscript reports a study of water treatment using high performance antifouling ultrafiltration polyether sulfone membranes incorporated with activated carbon. Specifically, the incorporation of APTMS modified AC were evaluated in terms of surface morphology, functional groups, mechanical properties, water contact angle, water retention, porosity, and water flux. Overall, this report presents a comprehensive work, although the FTIR may not be the best way to identify NH groups.

Reviewer 3 Report

Presented article shows the results of preparation and characterization of polyether sulfone ultrafiltration membranes containing modified APTMS activated carbon. Investigated filler and membranes were characterized by FTIR, SEM, contact angle, mechanical properties, antibacterial properties, water retention, porosity, and water flu. The presented results are interesting and worth to publish after major revisions and complements.

The remarks and comments are the following:

Abstract

Please change “angel” to “angle” – line 28.

Introduction

Please add information regarding the fillers that have been added to the PES matrix. In which filtration processes have PES been used as a membrane material?

Figure 9 and 10

On the figures and in the legend, please add the letters (A), (B), (C) etc.

Hydrophilicity of membranes

In order to check the hydrophilicity inside the membrane, the degree of swelling is measured. Please do these measurements.

Comparison with another membranes

Please compare obtained membranes with these reported in literature regarding the ultrafiltration process.

Round 2

Reviewer 3 Report

Presented revised article “Water Treatment Using High Performance Antifouling Ultrafiltration Polyether Sulfone Membranes Incorporated with Activated Carbon” shows the results of preparation and characterization of the polyether sulfone ultrafiltration membranes containing modified APTMS activated carbon. The authors took into account the reviewer's comments and added the relevant text, tables and figures to the manuscript. I suggest to accept the article in this form.
